# Potential for Peatland Water Table Depth Monitoring Using Sentinel-1 SAR Backscatter: Case Study of Forsinard Flows, Scotland, UK

Linda Toca [1,2,*], Rebekka R. E. Artz [2], Catherine Smart [2], Tristan Quaife [3], Keith Morrison [1], Alessandro Gimona [2], Robert Hughes [4], Mark H. Hancock [4] and Daniela Klein [4]

1 Department of Meteorology, University of Reading, Reading RG6 6BB, UK
2 The James Hutton Institute, Aberdeen AB15 8QH, UK
3 National Centre for Earth Observation, Department of Meteorology, University of Reading, Reading RG6 6BB, UK
4 Forsinard Flows RSPB Office, Forsinard KW13 6YT, UK
* Correspondence: linda.toca@hutton.ac.uk

**Abstract:** Peatland restoration has become a common land-use management practice in recent years, with the water table depth (WTD) being one of the key monitoring elements, where it is used as a proxy for various ecosystem functions. Regular, uninterrupted, and spatially representative WTD data in situ can be difficult to collect, and therefore, remotely sensed data offer an attractive alternative for landscape-scale monitoring. In this study, we illustrate the application of Sentinel-1 SAR backscatter for water table depth monitoring in near-natural and restored blanket bogs in the Flow Country of northern Scotland. Among the study sites, the near-natural peatlands presented the smallest fluctuations in the WTD (with depths typically between 0 and 15 cm) and had the most stable radar signal throughout the year (~3 to 4 dB amplitude). Previously drained and afforested peatlands undergoing restoration management were found to have higher WTD fluctuations (depths up to 35 cm), which were also reflected in higher shifts in the radar backscatter (up to a ~6 dB difference within a year). Sites where more advanced restoration methods have been applied, however, were associated with shallower water table depths and smoother surfaces. Three models—simple linear regression, multiple linear regression, and the random forest model—were evaluated for their potential to predict water table dynamics in peatlands using Sentinel-1 SAR backscatter. The random forest model was found to be the most suited, with the highest correlation scores, lowest RMSE values, and overall good temporal fit ($R^2$ = 0.66, RMSE = 2.1 cm), and multiple linear regression came in a close second ($R^2$ = 0.59, RMSE = 4.5 cm). The impact of standing water, terrain ruggedness, and the ridge and furrow aspect on the model correlation scores was tested but found not to have a statistically significant influence. We propose that this approach, using Sentinel-1 and random forest models to predict the WTD, has strong potential and should be tested in a wider range of peatland sites.

**Keywords:** Sentinel-1; peatlands; water table depth; SAR; peatland restoration

## 1. Introduction

Peatland ecosystems, in their natural state, both sequester $CO_2$ and act as long-term carbon storage. More than one-fifth of Scotland is covered by peatlands, making them a key landscape and a significant provider of peatland ecosystem services. Blanket bog is a unique habitat that requires very specific conditions to form, i.e., high rainfall, oceanic cool climate all year round, and relatively flat areas with poor surface drainage [1]. While quite common among UK peatlands, globally, these are rare habitats, covering only about 0.1% of the land area [2]. Inappropriate land management, such as drainage, commercial afforestation, agriculture, overgrazing, and peat cutting have left many peatlands around

the world degraded. Particularly in the UK, these poor past management decisions have turned peatlands from carbon sinks into large $CO_2$ emission sources. It was estimated that only about 20% of the UK's peatlands have remained in a near-natural state [3]. Luckily, more extensive landscape-scale peatland restoration has been observed in recent years [4].

For peatland restoration to be successful, multiple criteria should be met. The return of a high water table, relatively even surface, and regrowth of vegetation are some of the most important factors [5,6]. A high water table depth (WTD) and saturated soil are crucial both to maintain a healthy bog environment in near-natural sites and to encourage recovery in peatland restoration sites. If rewetting has been successful, it encourages the recovery and regrowth of bog vegetation communities, improves the ecosystem's resilience, and can reduce $CO_2$ emissions and improve sequestration rates [6,7]. The average water table depth over (sub)decadal timescales in peatlands is closely related to carbon accumulation and greenhouse gas exchange [8], with an expected increase in $CO_2$ release after a significant and prolonged water table lowering [9]. While field-gathered water table depth measurements are invaluable in obtaining the precise depth of the water table, the installation of monitoring equipment, maintenance, and data gathering can be time- and resource-consuming and provide only point measurements. Remotely sensed data, on the other hand, have the potential for frequent hydrological condition monitoring covering large, spatially contiguous areas.

Previous studies have analysed the relationship between peatland surface moisture conditions and the water table depth with optical, thermal, and, less frequently, radar satellite data [10,11]. From studies utilising radar imagery, some of the most recent have been focused on peatland condition assessment based on the bulk movement of the peat using the coherent differential phase component of the signal [12,13], while most studies still focus on incoherent backscatter strength analysis. Some studies using backscatter strength for peatland studies have shown promising results, reaching a coefficient of determination of up to 0.93 [14], although most have reported moderate to weak relationships between the peatland hydrological state and the backscatter [15–17]. A recent high-resolution radar backscatter laboratory study demonstrated a high sensitivity of radar backscatter to hydrological patterns in a peatland ecosystem [18]. Statistical models that have been used for WTD estimates in wetlands using radar backscatter data mainly include linear regression [15,16,19,20] and random forest [14,21,22], but usually do not compare the performance and outcomes of different models at the same time.

It is partly the complex nature of the SAR signal, as well as the high heterogeneity among peatlands, that makes peatland hydrological monitoring using radar technology complicated. Additionally, very few studies have focussed on the comparison of statistical models referring to peatlands in different conditions. With an increasing number of peatland restoration sites and the necessity of regularly monitoring their condition, radar remote sensing has the potential to provide data on restoration trajectories and effectiveness at both high temporal and spatial resolutions, covering large areas. Sentinel-1 provides the advantage of obtaining data in cloudy weather and provides archival data dating back to 2014.

In this study, we evaluated whether multitemporal satellite radar imagery can be used to monitor the water table depth in near-natural sites and sites that are undergoing restoration. To determine whether there is a potential application of Sentinel-1 for peatland hydrological monitoring, the following objectives were set for this study:

1. Investigate the correlation between Sentinel-1 radar data and WTD in blanket bogs in the Flow Country of northern Scotland, ranging from intact and near-natural sites to sites damaged by past afforestation and drainage, where restoration has recently started.
2. Create and test models with different complexities for WTD estimation using Sentinel-1 data.
3. Characterise the effects of surface roughness, hydrological condition (characterized by WTD measurements), seasonality, and time of acquisition in the developed models.

## 2. Materials and Methods

### 2.1. Study Area

The Royal Society for the Protection of Birds (RSPB) Forsinard Flows reserve (58°23′N, 3°51′W) is located in Northern Scotland and is part of the 4000 km² Flow Country blanket bog [1]. Figure 1 shows the distribution of peat soils in the northern part of Scotland and the location of the chosen blanket bog research site. The nearest Met Office weather station (Kinbrace, Hatchery, ~20 km away) is situated at 103 m asl, has a mean annual (1991–2020) precipitation of 949 mm/year$^{-1}$, and max. and min. air temperatures of 11.7 °C and 3.6 °C, respectively.

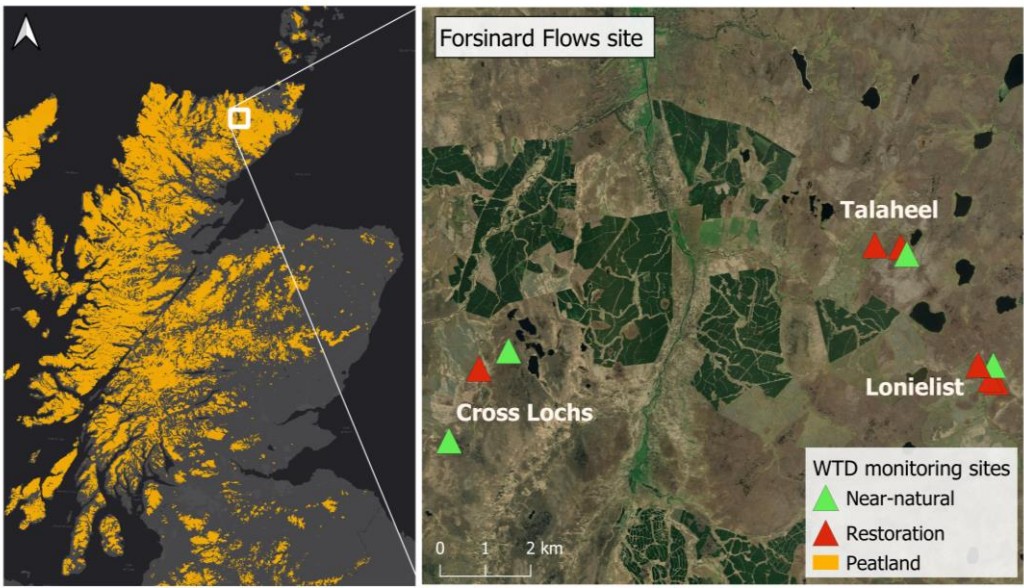

**Figure 1.** Peatland extent in northern Scotland [23] and the chosen study sites in the Forsinard Flows. The green triangles show the near-natural peatland study sites, and the red triangles show sites with ongoing restoration.

Technological forestry advancements, along with government tax incentives in the 1970s and 1980s, led to the peatlands in this area being heavily drained, gouged, and planted with non-native conifers, namely, Lodgepole Pine *Pinus contorta* and Sitka Spruce *Picea sitchensis*. The restoration works have taken place at different times from 1998 to the present (-further details in Section 3.1) and include multiple restoration techniques that offer a valuable and unique set of restored peatland chronosequence data. The fell-to-waste (FTW) restoration method involves tree felling and placing the felled trees in nearby furrows or drainage ditches. In such sites, the historical ridge and furrow forestry pattern is still present after 20 years since the tree felling and is easily visible in remotely sensed data. Felling to waste is rarely practiced these days, as timber of larger sizes is now generally extracted, but it serves as a useful experimental treatment with respect to legacy issues. Additional management techniques applied after felling to waste (or, more recently, timber harvesting) can include furrow-blocking, mulching of the brash, stump flipping, and other surface reprofiling methods [7]. Brash-crushing and furrow-blocking treatments usually include the leftover brash material (tops and small-diameter branch wood) further crushing down and blocking the individual plough furrows created by forestry planting [24]. Cross-tracking includes surface ground smoothing using an excavator that passes over a peatland and with its weight eliminates or reduces the relict forestry ploughing pattern [24].

Eleven representative sites from near-natural and restoration sites were chosen based on the presence of water level monitoring equipment in three types of areas (Figure 2), with four representing near-natural (see "Control" in Figure 2), three representing simple restoration (FTW), and four representing complex restoration (Restoration+). The latter

had additional management applied after the initial felling treatment. The control sites were selected to have matching slopes and aspects to the relative restoration treatment areas in each of the three areas (Cross Lochs, Talaheel, and Lonielist), and an additional control near the location of a long-term greenhouse gas monitoring site at Cross Lochs [25]. This second near-natural control is surrounded by more small water bodies than the other control areas around the wider area, and it was included in order to assess the variability in the near-natural conditions more generally. At Lonielist, two restoration treatments were included, one site with brash-crushing and furrow-blocking treatments, and the other with further ground-smoothing treatment (cross-tracking).

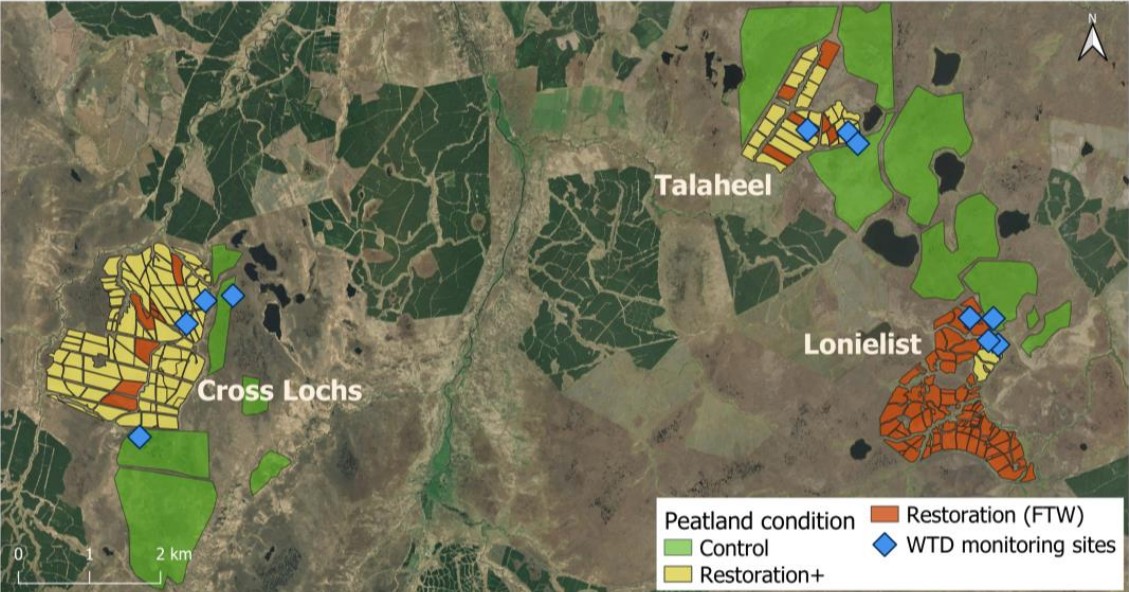

**Figure 2.** Forsinard Flows research site—different polygon colours represent the different management methods applied to peatland restoration. The eleven WTD monitoring points represent the chosen locations for analysis—4 near-natural Control sites, 3 Restoration (FTW), and 4 Restoration+ sites.

### 2.2. Water Table Depth and Meteorological Data

WTD data were recorded continuously from permanent dipwells using automatic loggers (Odyssey Capacitance Water Level Logger, New Zealand) at half-hourly intervals. The dipwells (~1.5 m length, 32 mm diameter) were made of polypropylene pipes and had 3 mm perforation holes at 50 mm intervals. Artz et al. [26] provide further details about the dipwell installation. The WTD data were combined to derive a daily average position (i.e., cm from the surface) for analysis to standardise with the Sentinel-1 overpasses. Water table depth data were available for the period between July 2017 (September 2018 for some sites) and October 2020. For mean WTD comparisons among the study sites, we used only the period when data for all sites were available (15 September 2018–31 July 2020).

### 2.3. Remotely Sensed Data

The Google Earth Engine (GEE) platform provides an invaluable computational environment for cloud-based processing of vast amounts of satellite imagery. GEE was used to acquire and process the Sentinel-1 high-resolution Level-1 Ground Range Detected (GRD) Interferometric Wide Swath (IW) products for the chosen study area and period.

GRD IW images were multi-looked and projected to ground range using an Earth ellipsoid model [27]. The GRD IW satellite imagery has a 20 × 22 m (range × azimuth) spatial resolution and 10 × 10 m pixel spacing. GRD scenes available on GEE were preprocessed, radiometrically calibrated, and corrected for terrain [28]. GEE was used to

extract satellite data between 2015 and 2021 for analysis at all 11 locations within Forsinard Flows. At each location, a 30 m radius around the water table depth monitoring station was considered, and therefore, each area analysed was equal to ca 2827 m$^2$ or about 28 Sentinel-1 pixels. A weighted reducer was used, where the pixels are included in the analysis if at least 50% of the pixel is in the region, and their weight is the fraction of the pixel covered by the region [28]. The 30 m distance was chosen to both include enough pixels to average out noise from the radar data and, at the same time, ensure the area analysed was relatively homogeneous. There was a stack of 330 Sentinel-1 images extracted for the chosen period, with values from both vertical–vertical (VV) and vertical–horizontal (VH) polarization. To secure data consistency, the data were then split into ascending and descending orbit datasets, and the ascending path was chosen as the most appropriate for the study region due to the ascending overpass happening around 6 pm in the evening, when the dew effect was expected to have a minor effect. The same path number (No. 30) was chosen for all images. The incidence angle difference for the areas investigated was small (ranging from 39.90° to 40.57°), and therefore, local incidence angle correction was not applied. As the extracted Sentinel-1 time series can be noisy, besides averaging the backscatter spatially over each of the locations, a temporal smoothing function with a rolling average (integer width of the rolling window (k) = 4) was applied to the dataset.

To improve the radar time-series data, weather-filtering can be beneficial [29] and was incorporated using data from the Centre for Environmental Data Analysis (CEDA) archive for the nearest long-term UK Met Office weather station, Altnaharra (~35 km away). Bechtold et al. [15] suggested the removal of days with heavy rainfall (>20 mm) or frozen soil (<2 °C), while Asmuß et al. [17] found the removal of dates where the soil temperature was below 2 °C and precipitation occurred in the six-hour period before acquisition improved the correlation coefficients between the grassland WTD and the radar backscatter. We found that the removal of days when the soil temperature was below 2 °C, days with snow cover, and days when rainfall heavier than 2 mm occurred in the 6 h period before the satellite overpass worked as the most optimal weather-filtering.

Additionally, the positioning of historical forestry ridge and furrow lines and the ratio of standing water at each location were visually derived from aerial photographs acquired over the area in 2017.

### 2.4. LiDAR Data and TRI Analysis

Surface roughness (microtopography and vegetation above the ground) is one of the parameters influencing radar backscatter. QGIS software was used to process LiDAR imagery, generate a digital surface model (DSM), and derive terrain ruggedness index (TRI) values. The TRI describes the amount of elevation difference between the cells of a DSM and was used as an indicator to analyse the surface roughness of the research sites. The TRI is calculated as an elevation difference between each DSM grid cell (50 × 50 cm) and its eight surrounding cells in a DEM using an equation developed by [30]:

$$\mathrm{TRI} = \sqrt{[\Sigma\,(x_i - x_0)^2]} \tag{1}$$

where $x_i$ is the elevation (m) of each neighbouring cell to the central cell $x_0$ (m).

### 2.5. Model Generation and Statistical Analysis

To examine the backscatter trends and the potential for Sentinel-1 backscatter to be used as a predictor of WTD for select peatland sites, three models with different complexities were chosen for the analysis: a simple linear regression (SLR), multiple linear regressions (MLR), and the random forest method (RF). The modelling dataset was split into training (70%) and validation (30%) datasets. The correlation coefficient ($R^2$) and associated *p*-values and the root mean square error (RMSE) were calculated and reported for all three modelling approaches. All statistical analyses were conducted in the R programming environment [31].

Some studies have shown values from a single polarization to be sensitive to water table depth changes [16,18,32]. Therefore, first, Sentinel-1 backscatter (VV and VH polarization, separately) was directly used as a predictor variable to model the water table depth using a simple linear regression model:

$$y = \alpha_0 + \beta_1 x_1 \tag{2}$$

where y is the dependent variable that is being predicted (WTD), $x_1$ is the independent variable (VV or VH polarization backscatter) used for the prediction of y, $\alpha_0$ is the intercept, and $\beta_1$ is the slope.

The model was then expanded to multiple linear regression, which can incorporate multiple continuous or categorical independent variables. The WTD was estimated using radar VV and VH polarizations and four categorical variables: the season and year of the radar acquisition, the site being investigated (site identifier), and the condition group it belongs to (Control/Restoration (FTW)/Restoration+).

$$y = \alpha_0 + \beta_1 x_1 + \beta_2 x_2 + \ldots + \beta_n x_n \tag{3}$$

where y is the dependent variable that is being predicted (WTD), $x_1, x_2, \ldots, x_n$ are the independent variables (VV, VH polarizations, season, year, site identifier, and condition group) used to predict y, $\alpha_0$ is the intercept, and $\beta_1, \beta_2, \ldots, \beta_n$ are the slope coefficients for each explanatory variable. The VV and VH variables were standardised using their z-score to compare the importance of each polarization coefficient for the model. The R stats package and lm function were used for SLR and MLR fitting [31].

Finally, a random forest (RF) machine learning approach was used to estimate the water table depth using the R randomForest package (version 4.7-1.1) [33]. Random forest, developed by Breiman [34], is a supervised learning algorithm that uses the ensemble method by grouping multiple decision tree predictions to perform regression or classification. One of the best-known RF advantages is the model's ability to identify non-linear relationships, as the model does not make assumptions about the relationship between the input and response variables [34]. The same input variables as those for MLR were used. The model was restricted to 50 decision trees and 3 variables to randomly sample as candidates at each split. The RF model feature importance was then reported, quantifying the relative importance of the different input variables for the WTD estimate.

## 3. Results

First, we examine the temporal dynamics of the observed water table depth and radar remote sensing data from all the sites. Then, the dynamics of the observed water table depth and the backscatter among the different condition groups are described. Finally, the model outcomes for the three different modelling approaches are reported and compared to the field observed water table depth data.

### 3.1. Water Table Depth and Sentinel-1 Time Series Analysis

A summary of the Forsinard Flows peatland sites' historical management, aerial photographs, calculated mean observed WTD (mm below the surface), and TRI are presented in Table 1 (Talaheel site), Table 2 (Lonielist site), and Table 3 (Cross Lochs site).

The summary presented in Tables 1–3 shows that the near-natural sites at all locations exhibited the desirable hydrological conditions for peatlands of a mean water table depth close to the surface. The WTD at the restoration sites with simplistic restoration presented the deepest values, while the WTD at sites with additional restoration measures were typically between that of the control sites and the felled-to-waste sites. Similarly, the TRI values were the lowest for the near-natural sites and the highest for the felled-to-waste sites, affected by the ridge and furrow patterns.

**Table 1.** Forsinard Flows Talaheel site's characteristics. * Mean water table depth between 15 September 2018 and 31 July 2020.

| | Near-Natural (Control) | Restoration (Felled to Waste) | Restoration+ (Felling and Additional Management) |
|---|---|---|---|
| Aerial photograph (natural colour (RGB) orthophoto, 50 cm resolution, March 2017) |  |  |  |
| Terrain ruggedness index, TRI (cm) | 9.6 | 21.3 | 12.5 |
| Mean WTD * ± SD (cm) | 4.9 ± 3.5 | 12.3 ± 4.0 | 11.3 ± 5.4 |
| Year afforested | - | 1983 | 1985 |
| Year felled | - | 1998 | 1998 |
| Year of additional restoration management | - | - | 2016 |
| Restoration method | - | - | Brash-crushing and furrow-blocking |
| Abbreviation | TA_CON | TA_FTW | TA_BCFB |

**Table 2.** Forsinard Flows Lonielist sites' characteristics. Two Lonielist restoration sites with additional management were used in this study and are described as (a) LO_BCFB and (b) LO_CT. * Mean water table depth between 15 September 2018 and 31 July 2020.

| | Near-Natural (Control) | Restoration (Felled to Waste) | Restoration+ (Felling and Additional Management) |
|---|---|---|---|
| Aerial photograph (natural colour (RGB) orthophoto, 50 cm resolution, March 2017) |  |  | (a)  (b)  |
| Terrain ruggedness index, TRI (cm) | 9.7 | 16.8 | (a) 14.0 (b) 17.0 |
| Mean WTD * ± SD (cm) | 2.8 ± 3.8 | 12.9 ± 5.6 | (a) 3.8 ± 4.7 (b) 9.4 ± 7.0 |
| Year afforested | - | 1985 | 1985 |
| Year felled | - | 2006 | (a) 2006 (b) 2004 |

**Table 2.** *Cont.*

|  | Near-Natural (Control) | Restoration (Felled to Waste) | Restoration+ (Felling and Additional Management) |
|---|---|---|---|
| Year of additional restoration management | - | - | (a) 2012, 2018<br>(b) - |
| Restoration method | - | - | (a) Brash-crushing and furrow-blocking<br>(b) Cross-tracking and furrow-blocking |
| Abbreviation | LO_CON | LO_FTW | (a) LO_BCFB<br>(b) LO_CT |

**Table 3.** Forsinard Flows Cross Lochs sites' characteristics. Two near-natural Cross Lochs sites were used in this study and are described as (a) CL_CON and (b) CL_CON_EC (location of the long-term GHG monitoring station). * Mean water table depth between 15 September 2018 and 31 July 2020.

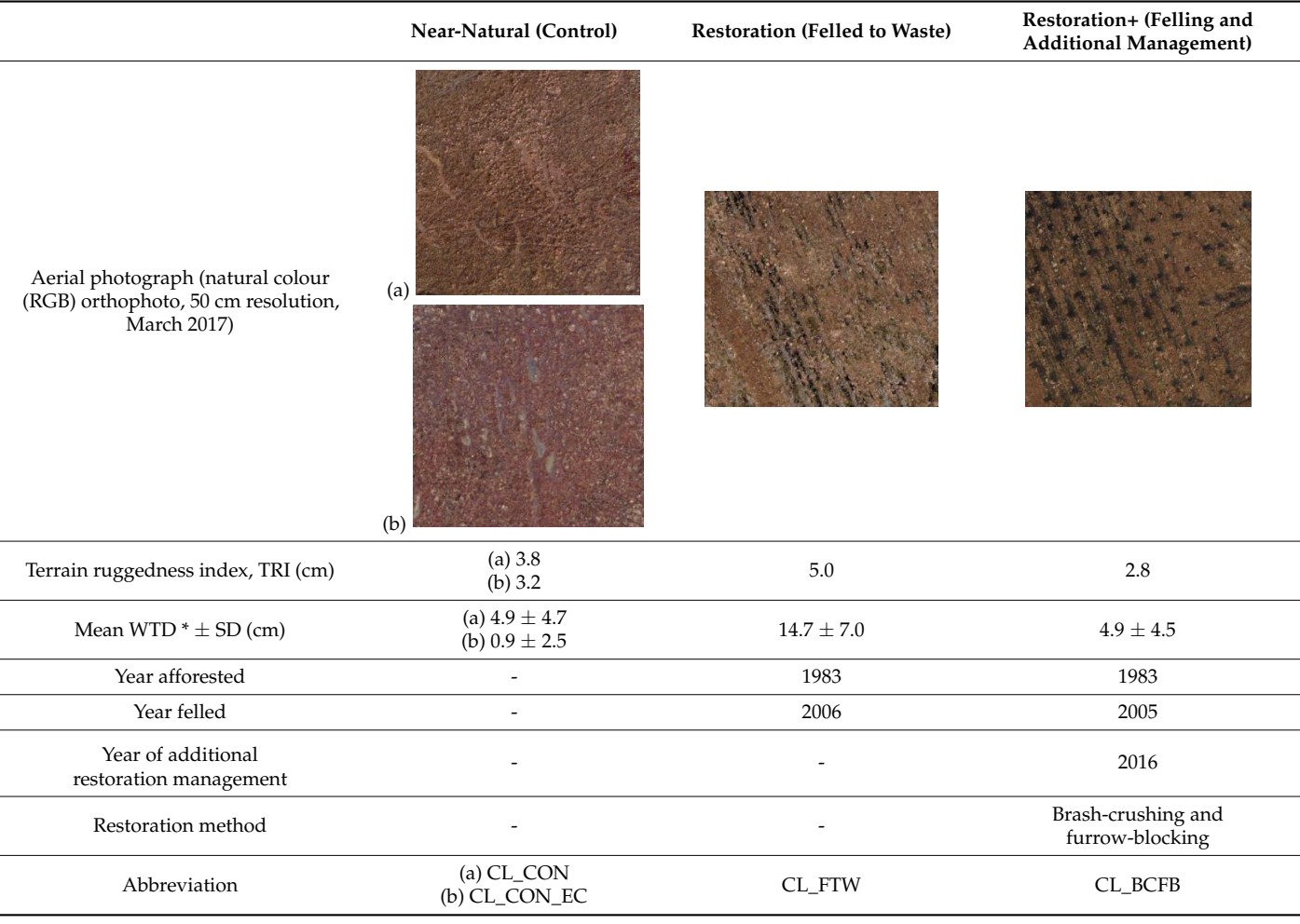

|  | Near-Natural (Control) | Restoration (Felled to Waste) | Restoration+ (Felling and Additional Management) |
|---|---|---|---|
| Aerial photograph (natural colour (RGB) orthophoto, 50 cm resolution, March 2017) | (a)<br>(b) | | |
| Terrain ruggedness index, TRI (cm) | (a) 3.8<br>(b) 3.2 | 5.0 | 2.8 |
| Mean WTD * ± SD (cm) | (a) 4.9 ± 4.7<br>(b) 0.9 ± 2.5 | 14.7 ± 7.0 | 4.9 ± 4.5 |
| Year afforested | - | 1983 | 1983 |
| Year felled | - | 2006 | 2005 |
| Year of additional restoration management | - | - | 2016 |
| Restoration method | - | - | Brash-crushing and furrow-blocking |
| Abbreviation | (a) CL_CON<br>(b) CL_CON_EC | CL_FTW | CL_BCFB |

### 3.1.1. Near-Natural Peatlands—Control Areas

Figure 3 shows the water table depth and Sentinel-1 time series for four near-natural peatland areas in the Forsinard Flows site. The water table depth (apart from the short periods during the 2019 and 2020 summer seasons) largely remained very close to the surface or inundated for long periods of the year. The mean water table depths of the sites were 4.9 ± 3.5 cm, 2.8 ± 3.8 cm, 4.9 ± 4.7 cm, and 0.9 ± 2.5 cm for the Talaheel, Lonielist, Cross Lochs, and Cross Lochs EC sites, respectively. While excluded from the mean WTD calculation, the Cross Lochs EC site suggested a significant WTD drawdown during

summer 2018, when data from the other sites were not available yet. VV polarization for all control sites was, on average, 8 dB stronger compared to the VH polarization. Among the control sites, the difference in the mean backscatter values was only about 1.5 dB between the sites with the strongest and weakest returns. The highest backscatter values were observed at the Cross Lochs (−19.6 dB VH; −11.7 dB VV) and Cross Lochs EC sites (−19.9 dB VH; −11.6 dB VV), followed by Lonielist (−20.6 dB VH; −12.3 dB VV), and the lowest values were found for Talaheel (−21.0 dB VH; -13.1 dB VV). As expected for non-forested bog ecosystems, all near-natural sites had very low TRI values (~3–4 cm for the Cross Lochs sites and ~10 cm for the Talaheel and Lonielist sites).

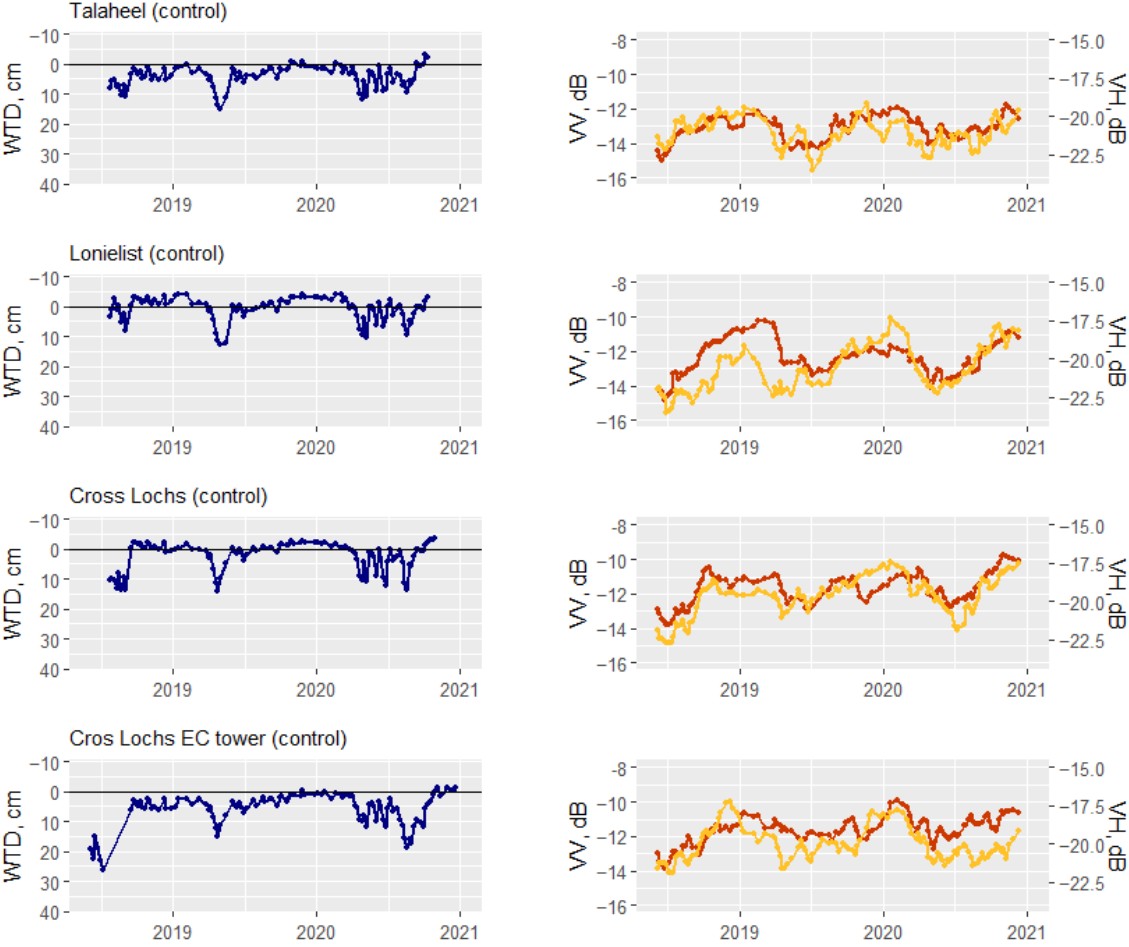

**Figure 3.** Observed water table depth (**left**) and Sentinel-1 ((**right**); VV in red, VH in yellow) backscatter time series for near-natural peatland sites in Forsinard Flows area. The black horizontal line indicates the ground surface.

### 3.1.2. Restoration—Felled to Waste

Figure 4 shows the water table depth and Sentinel-1 time series for three peatland restoration areas that have had a simple restoration technique applied—"felled to waste". The water table depth levels were lower compared to the previously analysed near-natural sites, and the summer drought periods led to deeper water levels and longer recovery periods. The mean water table depths of the sites were 12.3 ± 4.0 cm, 12.9 ± 5.6 cm, and 14.7 ± 7.0 cm for the Talaheel, Lonielist, and Cross Lochs sites, respectively, and none of the loggers were inundated throughout the measuring period. As with the near-natural sites, the Cross Lochs data series, which date further back, indicate a water table depth drawdown in the summer of 2018. The highest backscatter values were observed at Cross Lochs (−19.7 dB VH; −11.6 dB VV), followed by Lonielist (−20.3 dB VH; −12.7 dB VV), and Talaheel (−20.0 dB VH; −13.3 dB VV). The VV backscatter values were, on average,

0.4 dB weaker compared to the near-natural sites, while the VH backscatter was 0.3 dB stronger. The TRI values were found to be the highest for Talaheel with ~21 cm and were around ~17 cm for Lonielist and ~5 cm for the Cross Lochs site.

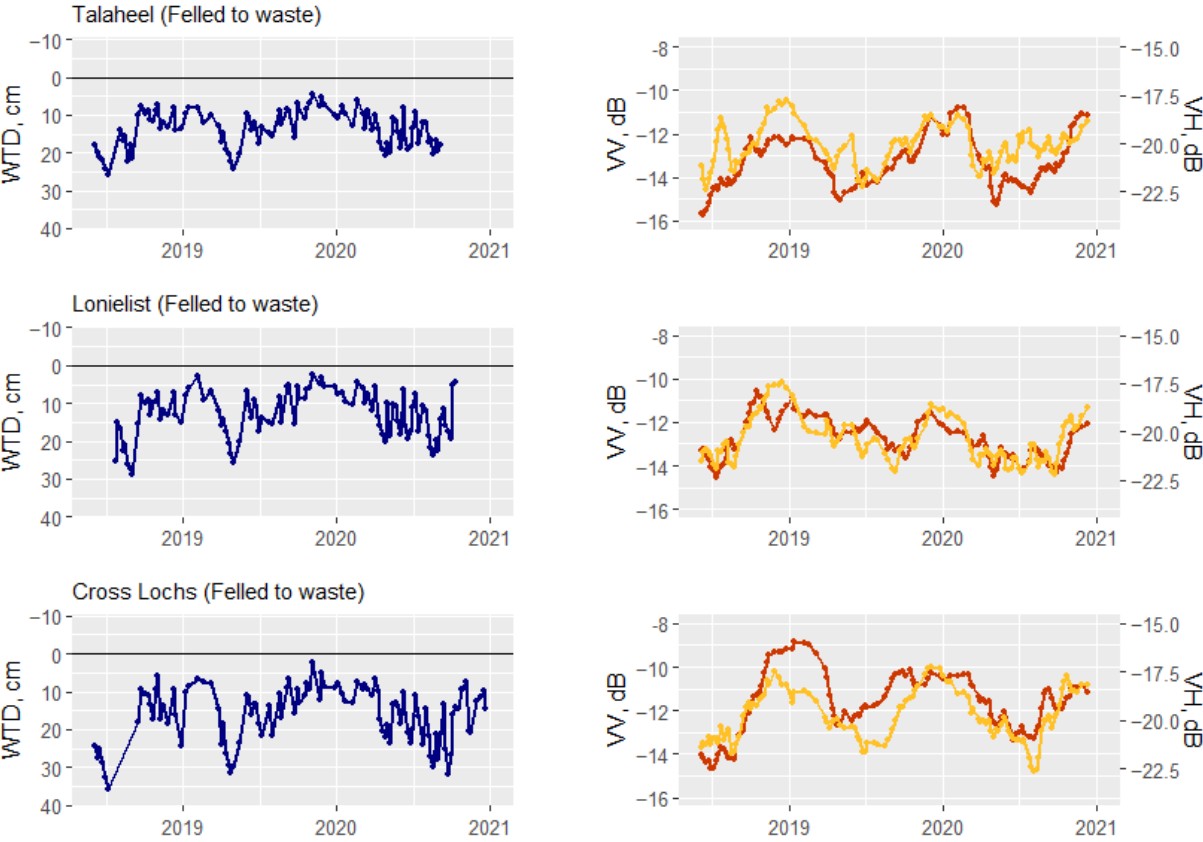

**Figure 4.** Observed water table depth (**left**) and Sentinel-1 ((**right**); VV in red, VH in yellow) backscatter time series for peatland sites with simplistic restoration (felling to waste) applied. The black horizontal line indicates the ground surface.

### 3.1.3. Restoration with Additional Management

Figure 5 shows the water table depth and Sentinel-1 time series for four peatland restoration areas where, in addition to felling, other potentially more successful peatland restoration methods have been used. The additional methods for these sites included furrow-blocking, brash-crushing, and cross-tracking. The time series of the water table depth at these sites greatly resembled those observed at the near-natural sites, with more stable water levels, smaller drawdowns, and quicker recoveries during the dry seasons. Partial inundation was also observed, most notably at the Cross Lochs site. The mean water table depths of the sites were $11.3 \pm 5.4$ cm, $3.8 \pm 4.7$ cm, $9.4 \pm 7.0$ cm, and $4.9 \pm 4.5$ cm for the Talaheel, Lonielist-1, Lonielist-2, and Cross Lochs sites, respectively. The highest backscatter values were observed at the Lonielist cross-tracked site ($-18.0$ dB VH; $-10.3$ dB VV), followed by the Talaheel ($-19.5$ dB VH; $-12.6$ dB VV), Cross Lochs ($-19.9$ dB VH; $-12.3$ dB VV), and Lonielist ($-19.9$ dB VH; $-12.4$ dB VV) brash-crushed and furrow-blocked sites. The TRI values were found to be slightly lower compared to the felled-only sites, being the highest for Lonielist (14–17 cm), followed by Talaheel (~12 cm), and the lowest for the Cross Lochs site (~3 cm).

The average radar backscatter values for this group were found to be the strongest among the three condition groups. The VH backscatter was 1 dB stronger than the control sites and 0.7 dB stronger than the felled-to-waste sites. The VV backscatter was slightly stronger (0.3 dB) compared to the control sites and 0.6 dB stronger than the felled-to-waste sites.

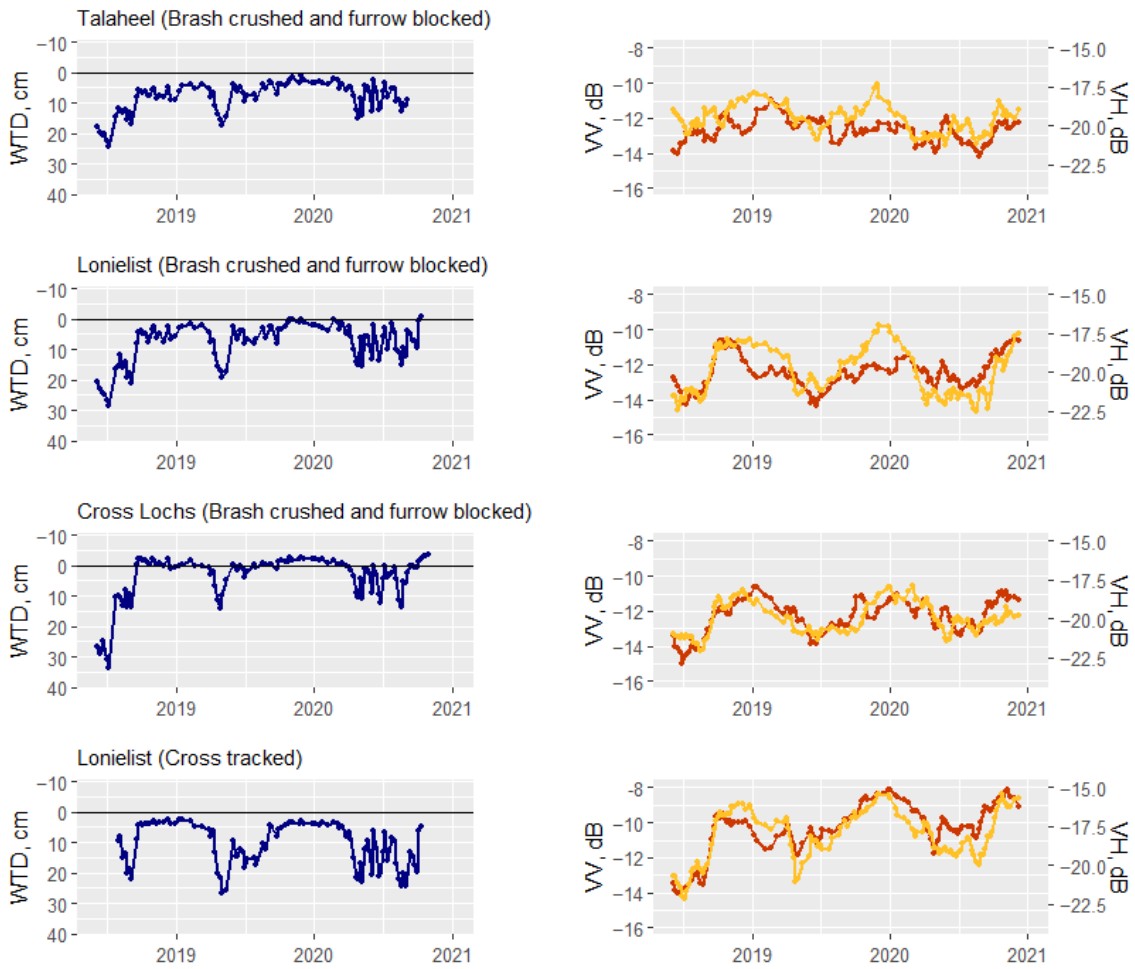

**Figure 5.** Observed water table depth (**left**) and Sentinel-1 ((**right**); VV in red, VH in yellow) time series for sites undergoing more advanced restoration applied (brash-crushing, furrow-blocking, and re-profiling). The black horizontal line indicates the ground surface.

### 3.2. Correlation of Sentinel-1 Backscatter and Water Table Depth

#### 3.2.1. SLR Model

First, the SLR model was applied to all sites and condition groups together. A very low agreement between the predicted and observed WTDs was found ($R^2 < 0.01$ when using either VV or VH) (Figure 6a and Table 4 in Section 3.2.3). When the model was applied to each site separately, an improved but still very weak relationship ($R^2 < 0.3$) was found for all sites except the Talaheel Restoration (FTW) site ($R^2 = 0.47$) (Figure 6b). An evident clustering of the condition groups is visible in Figure 6a, but the model was not able to predict any WTD below 9 cm depth. VV was found to be a slightly stronger predictor compared to VH polarization. Given that in some sites, such as the near-natural ones, the WTD during the year only fluctuated within about the 15 cm range, the RMSE was found to be quite high (RMSE = 7.1 cm for the training data and 7.4 cm for the validation dataset).

#### 3.2.2. MLR Model

A multiple linear regression (MLR) model was fitted to estimate the WTD using VV and VH polarization backscatter and four categorical variables: season and year of radar image acquisition, area investigated, and the condition group it belongs to. For the combined data input, the MLR model yielded $R^2 = 0.59$, with very similar scores for all condition groups ($R^2 = 0.46$ for the Control and Restoration (FTW) groups and $R^2 = 0.44$ for the Restoration+ group) (Figure 7a and Table 4 in Section 3.2.3). When applied to sites individually, moderate agreement between the observed and predicted WTD was found

at 2 sites ($R^2 > 0.5$), weak agreement ($0.3 < R^2 < 0.5$) at 8 sites, and very weak agreement ($R^2 < 0.3$) at 1 site (Figure 7b).

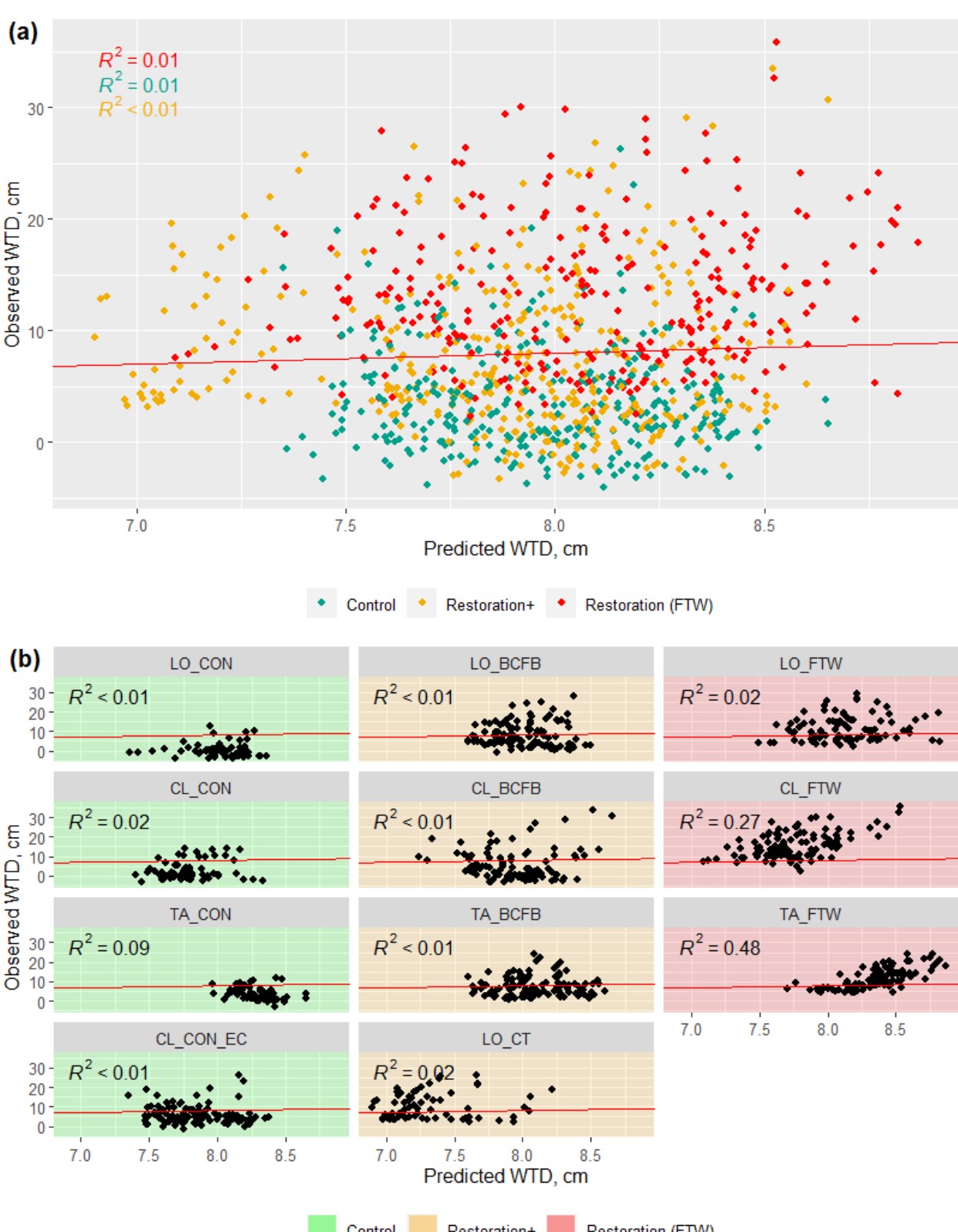

**Figure 6.** SLR model output for Forsinard Flows sites (predicted vs. observed WTD) for (**a**) all data points; (**b**) splitting data into the individual sites. The red line represents a perfect agreement between the prediction and the field measurement.

**Table 4.** SLR, MLR, and RF model performance using training (70%) and validation (30%) datasets, terrain ruggedness index (TRI), standing water presence (percentage of the total area), and forestry lines position (aspect in degrees, with True North being 0°) of individual sites. *** Indicates a 0.001 significance level; ** indicates a 0.01 significance level; * indicates a 0.05 significance level.

| Treatment | Site | Simple Linear Regression, SLR ($R^2$) | | Multiple Linear Regression MLR ($R^2$) | | Random Forest, RF ($R^2$) | | Terrain Ruggedness Index, TRI (cm) | Standing Water (%) | Forestry Ridge and Furrow Lines (Aspect, °) |
|---|---|---|---|---|---|---|---|---|---|---|
| | | Training | Validation | Training | Validation | Training | Validation | | | |
| Near-natural (Control) | TA_CON | 0.09 * | 0.02 | 0.45 *** | 0.26 ** | 0.30 *** | 0.25 ** | 9.6 | 2 | - |
| | CL_CON | 0.02 | <0.01 | 0.43 *** | 0.52 *** | 0.45 *** | 0.61 *** | 3.8 | 5 | - |
| | LO_CON | <0.01 | 0.02 | 0.20 *** | 0.25 *** | 0.16 *** | 0.21 ** | 9.7 | 5 | - |
| | CL_CON_EC | <0.01 | 0.04 | 0.36 *** | 0.52 ** | 0.43 *** | 0.52 *** | 3.2 | 15 | - |
| | Control group | 0.01 | <0.01 | 0.46 *** | 0.50 *** | 0.48 *** | 0.50 *** | 6.6 | | |
| Restored (felled to waste) | TA_FTW | 0.48 *** | 0.09 | 0.53 *** | 0.16 ** | 0.56 *** | 0.14 * | 21.3 | 5 | −60° |
| | CL_FTW | 0.27 *** | 0.08 | 0.36 *** | 0.19 ** | 0.36 *** | 0.18 ** | 5.0 | 5 | −26° |
| | LO_FTW | 0.02 | 0.04 | 0.36 *** | 0.61 *** | 0.29 *** | 0.35 *** | 16.8 | 20 | −17° |
| | Restoration (FTW) group | 0.01 | <0.01 | 0.46 *** | 0.34 *** | 0.44 *** | 0.26 *** | 14.4 | | |
| Restored with additional management | TA_BCFB | <0.01 | <0.01 | 0.44 *** | 0.21 ** | 0.53 *** | 0.06 | 12.5 | 45 | −16° |
| | CL_BCFB | <0.01 | 0.02 | 0.40 *** | 0.36 *** | 0.63 *** | 0.39 *** | 2.8 | 45 | −22° |
| | LO_BCFB | <0.01 | <0.01 | 0.52 *** | 0.43 *** | 0.54 *** | 0.51 *** | 14.0 | 30 | −30° |
| | LO_CT | 0.02 | <0.01 | 0.31 *** | 0.41 *** | 0.66 *** | 0.38 *** | 17.0 | 5 | −57° |
| | Restoration+ Group | <0.01 | 0.04 | 0.44 *** | 0.44 *** | 0.61 *** | 0.40 *** | 11.6 | | |
| Combined | | <0.01 | <0.01 | 0.59 *** | 0.62 *** | 0.66 *** | 0.60 *** | | | |

VH was ranked as a stronger predictor compared to VV. Similarly, autumn and winter values explained a greater amount of the unique variance over the spring/summer values. The year 2018 clearly stood out from the other years, coinciding with the 2018 European summer drought [35], and it had an exceptionally low water table depth in the spring and summer seasons. Compared to the SLR model, the RMSE improved and was within a 5 cm error (RMSE = 4.5 cm). When running the MLR model on the validation data, the correlation slightly increased, but the deviation of the residuals remained the same ($R^2$ = 0.62, RMSE = 4.5 cm).

Figure 8 shows the distribution of the residuals for the MLR model. It can be noticed that overfitting was more common, especially with a WTD below 12 cm; this is also the depth at which the residuals became more scattered. Overall, a normal distribution of the residuals was observed (see Normal Q-Q plot, Figure 8), but the model did not predict as well at the higher WTD ranges (lowest WTD) as it did for the lower ranges (WTD close to the surface). Therefore, even with the strong capillary connection occurring between the water table depth and the soil surface in peatlands [17,36], this indicates the limitations of the C-band's penetration ability and, subsequently, the predictive modelling capability for peatlands with deep mean annual water tables or during significant water table drawdown periods.

### 3.2.3. RF Model

Finally, the random forest model was applied, using the same input variables as for the MLR model. Good agreement ($R^2$ = 0.66) was found between the observed and predicted WTD when looking at the combined data, and out of the three condition groups, it was found to be highest for sites with more advanced restoration methods applied ($R^2$ = 0.48 for the Control, $R^2$ = 0.44 for the Restoration (FTW), and $R^2$ = 0.61 for the Restoration+ group) (Figure 9a and Table 4). When applied to sites individually, moderate agreement between the observed and predicted WTD was found at five sites

$(0.65 < R^2 < 0.5)$, with weak agreement $(0.3 < R^2 < 0.5)$ at four sites, and very weak agreement $(R^2 < 0.3)$ at two sites (Figure 9b). The RMSE score was further decreased compared to the two linear models, with RMSE = 2.1 cm (training data). When running the model on the validation data, the correlation slightly decreased, and the deviation of the residuals increased ($R^2$ = 0.60, RMSE = 4.2 cm). The variable ranking, starting with the most important, were the season, group, year, VH, site identifier, and VV (Figure 10).

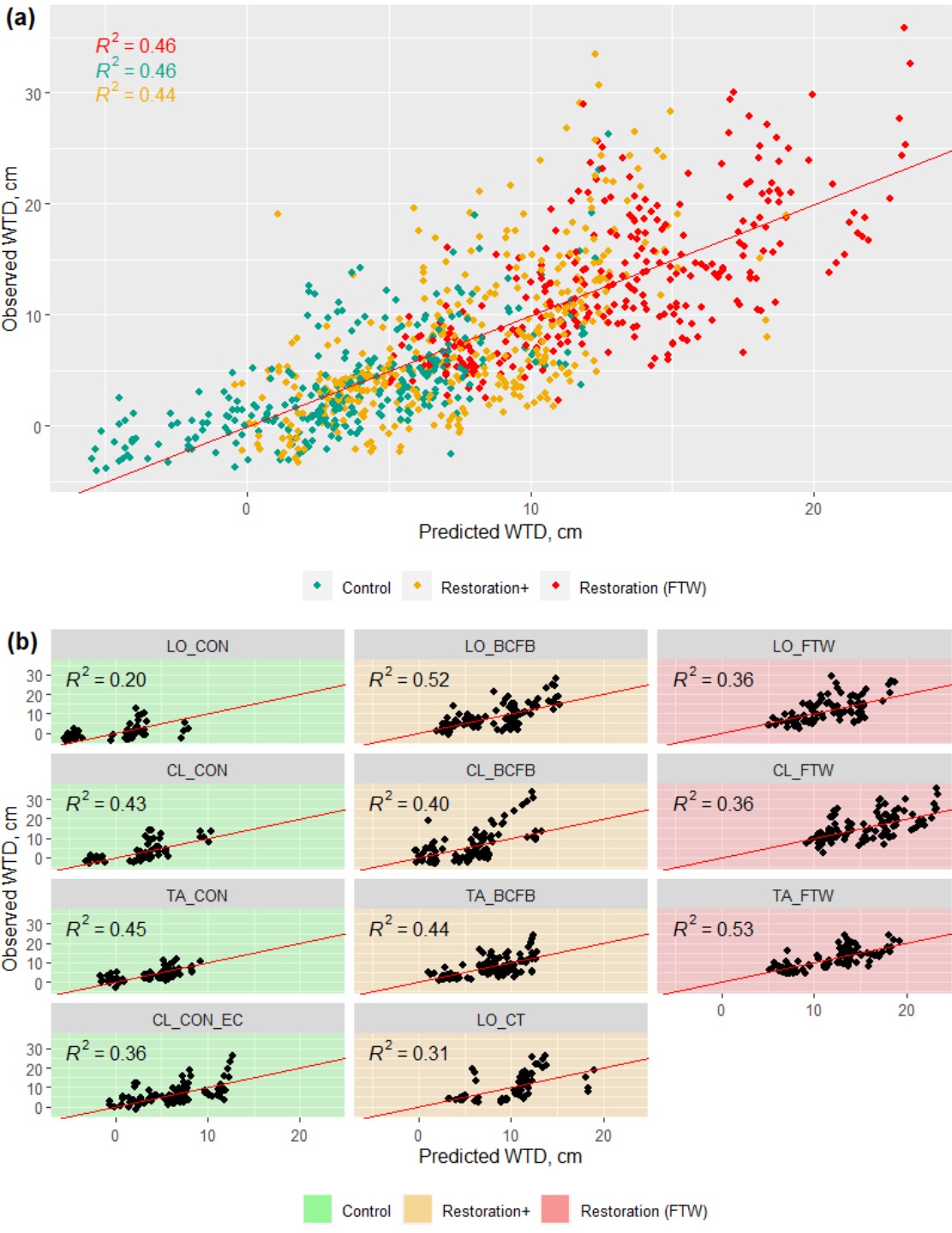

**Figure 7.** MLR model output for Forsinard Flows sites (predicted vs. observed WTD) for (**a**) all data points; (**b**) splitting data into the individual sites. The red line represents a perfect agreement between the prediction and the field measurement.

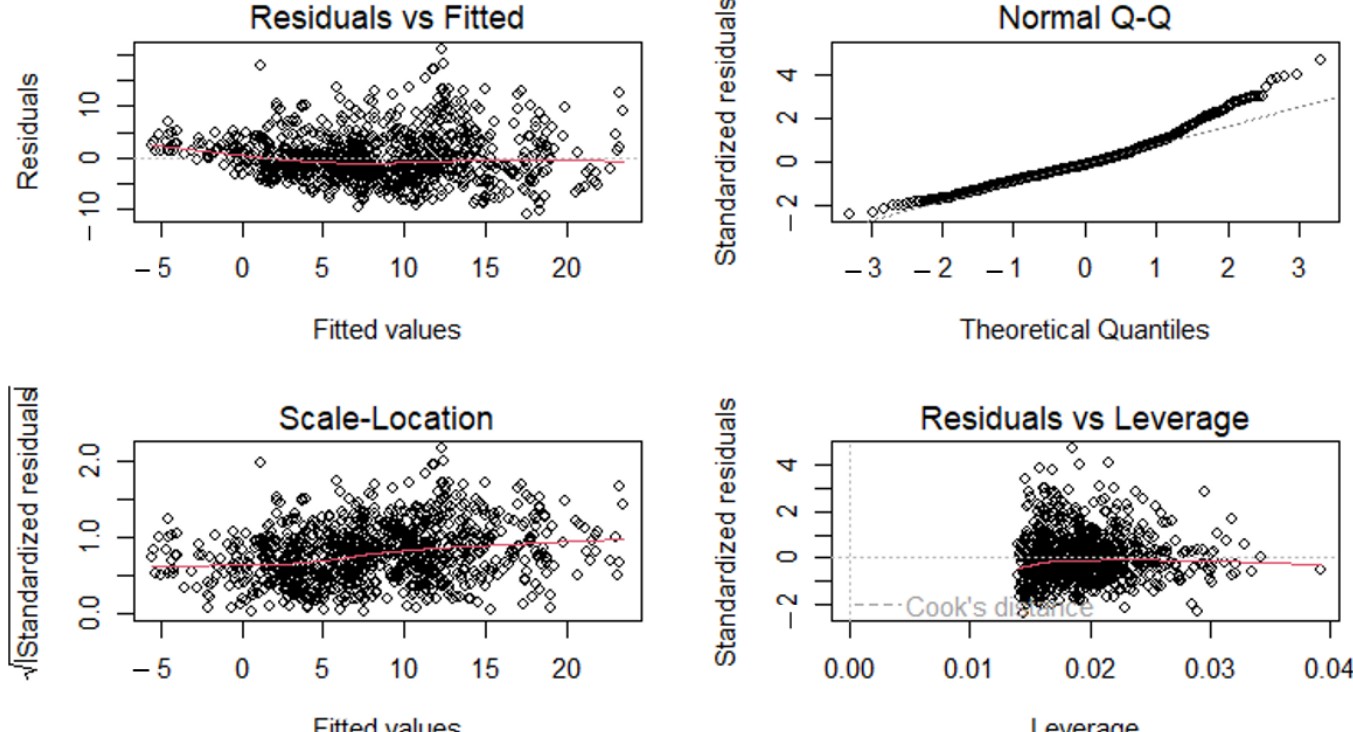

**Figure 8.** MLR model residual statistics. Clockwise from upper left: (1) Comparison of the residuals of the MLR model against the fitted values produced by the model, the red lines indicate a LOWESS fit; (2) quantile–quantile plot: confirms that both sets of quantiles came from the same (normal) distribution, except the upper quantiles, where the points stray above the line, indicating some deviation of the error from normality; this, however, is a small portion of the data; (3) scale–location plot: uses the square root of the standardized residuals instead of the residuals themselves; (4) leverage plot: no standardized residuals are outside of the Cook's distance boundaries, indicating that there were no strong outliers influencing the regression results.

The correlation coefficients in the three models, along with the TRI and standing water and forestry information for the individual sites, are collated in Table 4. The SLR model did not meet an acceptable predictive performance, and therefore, only the MLR and RF models are further discussed in detail.

The RF was found to have the highest $R^2$ and lowest RMSE values using the training dataset when looking at the combined data ($R^2$ = 0.66, RMSE = 2.1 cm for RF; $R^2$ = 0.59, RMSE = 4.5 cm for MLR). When using the RF model on the withheld validation dataset, the $R^2$ and RMSE decreased and had a similar performance to the MLR model ($R^2$ = 0.60 for RF, $R^2$ = 0.62 for MLR); however, the RMSE was still smaller for the RF model (RMSE = 4.2 cm for RF, RMSE = 4.5 cm for MLR). When investigating the sites individually, the RF was superior for 10 out of 11 sites investigated using the training data, while the MLR model was superior for 9 out of 11 sites when using the validation dataset. Among the peatland condition groups, the highest scores were found for the restoration sites with additional management applied (brash-crushing, furrow-blocking, or cross-tracking), followed by the control sites, and finally, the felled-to-waste sites. Figure 11 shows the MLR and RF models and the field observed water table depth series for all 11 sites, and in general, good agreement can be observed in terms of the temporal fit of the models. The typical problematic periods included the periods of water table drawdown, when the models systematically underestimated the water table depth.

Finally, the TRI, the proportion of standing water at each site, and the ridge and furrow aspect for the restoration sites were compared to the model performance scores by fitting a simple linear regression. None of the three variables reached a statistically significant level

($p$-value > 0.05), and all had very low $R^2$ scores ($R^2 = 0.07$ for the TRI; $R^2 = 0.2$ for standing water; $R^2 = 0.04$ for the ridge and furrow aspect). Hence, overall, none of these aspects had a strong influence on the models' performance.

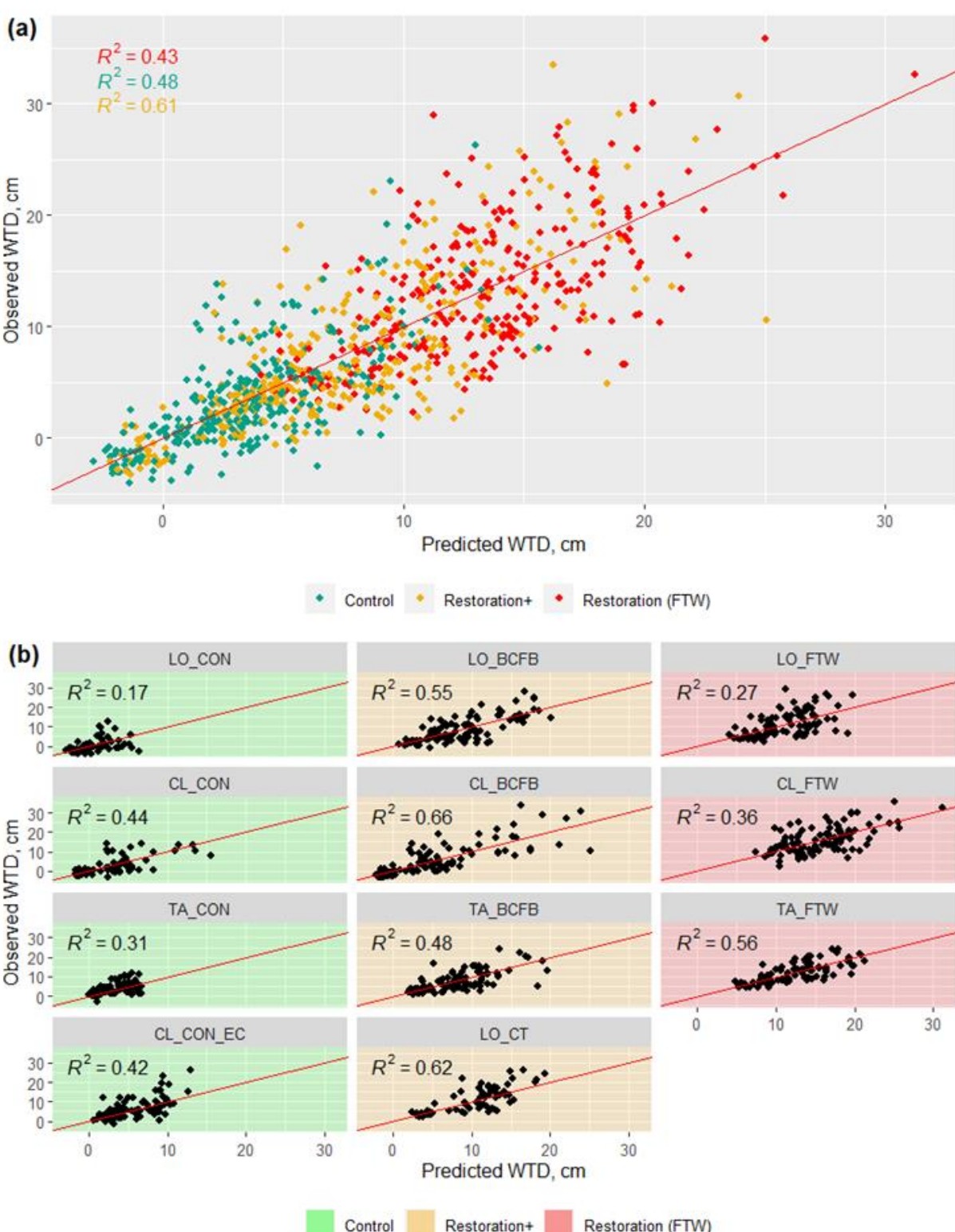

**Figure 9.** RF model output for Forsinard Flows sites (predicted vs. observed WTD) for (**a**) all data points and (**b**) splitting data into the individual sites. The red line represents a perfect agreement between the prediction and the field measurement.

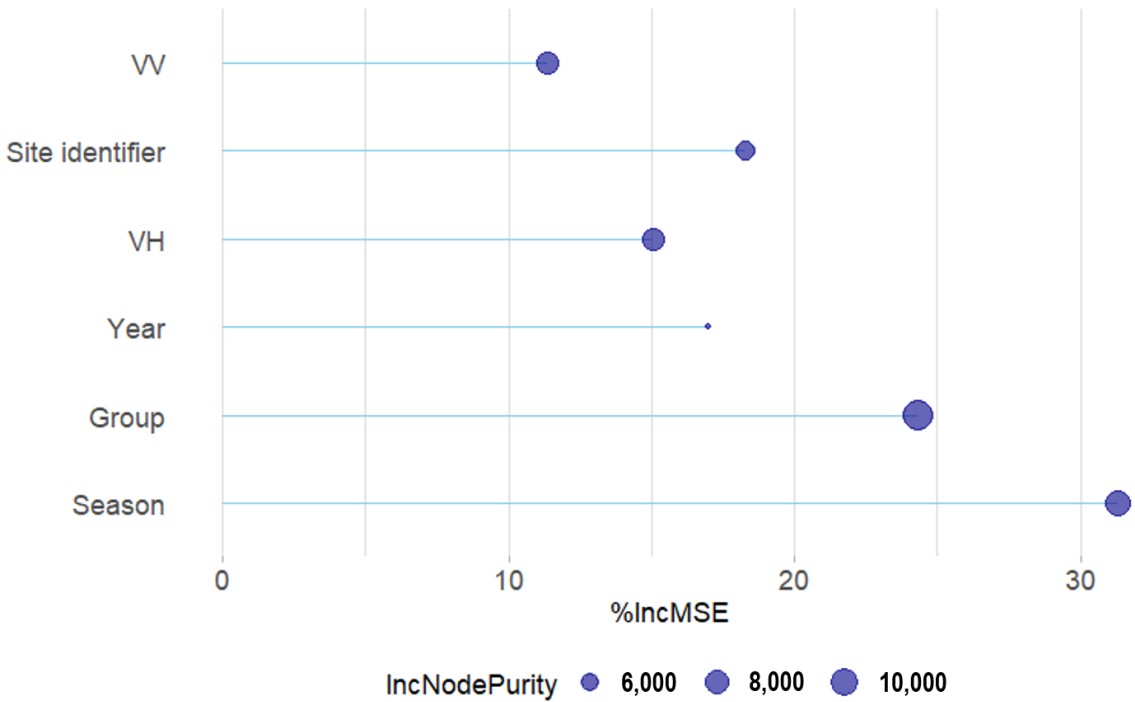

**Figure 10.** Random forest model variable importance for the WTD prediction. The percentage increase in the mean square error (%incMSE) is shown on the *x* axis, and the circles represent the increase in node purity. The smaller these two parameters, the less change observed in the model when a specific variable was removed or added. The variable ranking shows well how valuable the inclusion of categorical variables was for the improvement of the WTD prediction.

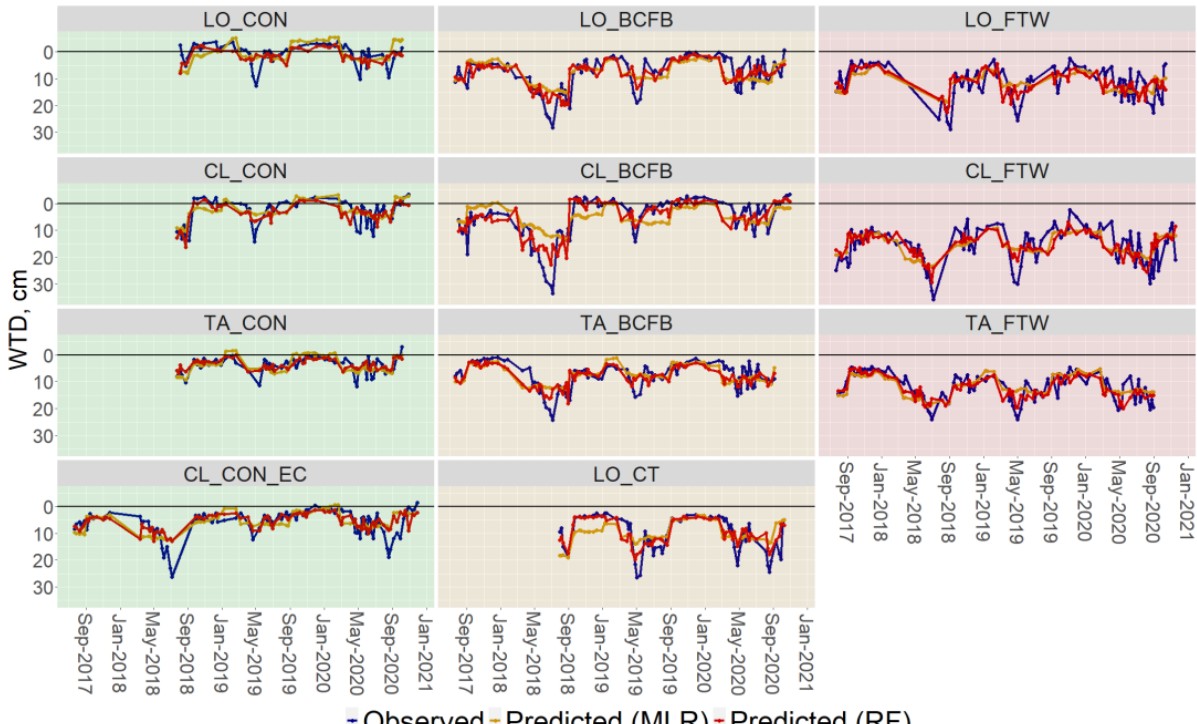

**Figure 11.** Modelled and observed water table depth series based on the MLR and RF models. The black horizontal line indicates the ground surface. Panels with a green background indicate the control sites, orange—Restoration+ sites, and red—Restoration (FTW) sites.

## 4. Discussion

This study illustrates the strong potential of multitemporal satellite radar imagery for modelling water table depth in near-natural and restored Scottish blanket bogs. If applied successfully, the remotely sensed radar time series could provide the opportunity for near-continuous monitoring of hydrological conditions, which would be especially valuable in extensive and remote peatlands, such as the Flow Country and many others. In this study, we investigated the dynamics of the water table depth and radar backscatter data in peatlands and compared simple linear regression (SLR), multiple linear regression (MLR), and random forest (RF) model applications to model the water table depth. Based on the $R^2$ and RMSE scores, the results show that the best-performing model for water table depth prediction was the RF, with the MLR model being a close second, while the SLR model did not meet acceptable predictive performance. An independent validation was performed for all three models using the withheld validation dataset, and the model performance results were similar to the outcomes of the training dataset (a 6% decrease in the $R^2$ for the RF and a 3% increase for the MLR). In addition to the VV and VH radar backscatter variables, the inclusion of categorical variables (year and season of image acquisition, and the site and condition group being investigated) significantly improved the model scores, reaching up to 66% of the WTD variance explained. If these covariables were removed from the input, the model score decreased to 54% using the RF, while the MLR performance was so low that it was no longer useful for WTD prediction. The model scores varied significantly among the individual sites ($R^2$ = 0.16–0.61 for MLR; $R^2$ = 0.06–0.66 for RF) indicating high heterogeneity, which could not be fully explained solely by the variation in the terrain ruggedness, the presence of standing water, or the ridge and furrow aspect. Indeed, peatland ecosystems can exhibit a high degree of sub-pixel heterogeneity in vegetation composition and microtopography [1] within the 20 × 22 m spatial resolution of Sentinel-1 imagery. As the Sentinel-1 imagery pixel size is smaller than the products' spatial resolution, adjacent pixels are correlated and resulting backscatter values can be impacted by the surrounding area. Räsänen et al. [22], in their study, concluded that the high variation in the correlation among the sites is often explained by the heterogeneity among them, which seems to support the results from this study. Scholefield et al. [37] found that the inclusion of a topography variable in their random forest model for peatland habitat extent increased the model classification performance. Future studies on water table depth modelling from satellite radar data should investigate if more detailed surface topography (aspect, slope, and topographic index) could improve the WTD estimates. In our study sites, some of the specific features that cause heterogeneity among the sites and, consequently, could have influenced the WTD–backscatter relationship are:

1. Topography and microtopography: gullies, hags, hummocks, hollows, pools, ridge and furrow pattern;
2. Soil and vegetation moisture content, inundation;
3. Varying vegetation;
4. Soil density and texture.

Of our study sites, the near-natural sites experienced the smallest fluctuation in the water table depth and, consequently, the radar signal was most stable throughout the year in these areas. In agreement with the findings from the study by Holden et al. [38], we found previously drained peatlands to have higher fluctuations even years after restoration, which caused the radar backscatter to experience higher shifts throughout the year. Higher backscatter values were typically observed in the autumn and winter season when the water table depth was typically closer to the surface, and lower backscatter values in spring and summer when the WTD dropped. Seasonal patterns were also observed in the water table depth–backscatter relationship, and therefore, the decision to include the season as a variable in the model was made. We found that the inclusion of the radar image acquisition time (year and season) improved the WTD prediction by 25% using RF and by 19% using MLR.

For some sites, we hypothesised that standing water, inundation, and a water table depth that remains close to the surface could be the reasons for the low relationship scores. This was particularly noticeable when using the SLR model for the Restoration+ sites, such as at Talaheel and Cross Lochs. However, when using the MLR and RF models, the $R^2$ scores for these sites significantly increased. Both Bechtold et al. [15] and Lees et al. [39] have noted how the WTD from the near-natural peatland sites can have low agreement with SAR data due to the low fluctuations throughout the year. Asmuß et al. [17], in their study on grasslands with organic soils, completely excluded areas where the WTD was shallower than 5 cm, as the ground surface can be partly inundated in these instances, leading to a contrary relationship between the radar backscatter and the WTD. For our study sites in the Forsinard Flows, excluding the 2018 drought year, the mean annual WTD in the four-year monitoring period only exceeded a 10 cm depth at four sites: Cross Lochs Restoration (FTW) ($14.7 \pm 7.0$ cm), Talaheel Restoration (FTW) ($12.3 \pm 4.0$ cm), Lonielist Restoration (FTW) ($12.9 \pm 5.6$ cm), and Talaheel Restoration+ ($11.3 \pm 5.4$ cm). It was expected that these slightly drier sites might have had higher correlation scores given the higher water table depth fluctuations and the low presence of standing water. However, the models yielded only moderate to low scores for these sites. The presence of standing water at the sites, along with the historical ridge and furrow patterns, was expected to have an influence on the correlation scores given the strong impact on radar backscatter of specular and rugged surfaces. There was no significant relationship found between the percentage of standing water and the modelling outcome. This, however, was a relatively small sample (11 sites) for testing the influence of such a parameter, so further studies, where repeated annual and season data on open water presence in the research areas are available, could be beneficial to investigate the seasonal backscatter–WTD relationships. Similarly, only seven sites with previous afforestation were tested for the influence of the ridge and furrow pattern. In ploughed fields where the ridges are oriented perpendicular to the radar signal, this can result in stronger backscatter, whereas, if they are parallel to the radar signal, they may not affect the backscatter as much [40]. From the Sentinel-1 backscatter time series, we could not see a pattern where the sites with ridges more parallel to the radar signal would be weaker (e.g., the Talaheel felled-to-waste site or the Cross Lochs cross-tracked site), or the opposite for more perpendicular sites (e.g., Talaheel brash-crushed and furrow-blocked site), nor was there a statistically significant relationship found between the aspects and the three model's $R^2$ scores. It is worth mentioning that these are restoration sites covered with peatland vegetation, so the impact of the ridge and furrow pattern is expected to be much lower than that of a freshly ploughed agricultural field, where this effect is known to have a stronger impact. A study using a larger dataset and focussing on the topographical (ridge and furrow line) aspect impact on radar backscatter in previously afforested peatlands would be valuable for further modelling improvements.

Overall, blanket bogs dominated by mosses, sedges, heath, and heather experience small changes in vegetation throughout the year. *Sphagnum* and other mosses do not die back during winter and have the ability for year-round growth even with snow cover [41]. Heather *Calluna vulgaris* does not die back during winter either; however, the leaves and flowers lose their colour and turn browner. Similarly, deergrass *Trichophorum cespitosum* and cotton grasses *Eriophorum* sp. turn from green to rusty brown, with deergrass eventually dying back by late winter.

In this study, we did not apply any explicit correction to the microwave signal to account for vegetation. This may account, in part, for the lower prediction skill of the simpler models. Some previous studies have included a vegetation sine correction Equation (4) in the Sentinel-1 data processing to account for the growing season vegetation in peatlands in the Forsinard Flows reserve [39] and Finnish peatlands [22].

$$\sigma_v = \sigma - \sin(0.0173 \times (\text{DoY} - a)), \tag{4}$$

where σ$_v$ is the sine-corrected radar backscatter (dB), σ is the backscatter before vegetation correction (dB), DoY is the day of the year of the radar image acquisition, and *a* is the approximate day the growing season begins.

The correction in these studies was based on the day of the year of the radar image acquisition and the approximate day when the growing season begins. We found that accounting for the seasonal greening up of vegetation with a sine function can alter the data in an inappropriate manner by inflating the backscatter values, and therefore, also the model fit (see Supplementary Figure S1). More advanced vegetation corrections have been proposed for Sentinel-1 data normalization to account for the vegetation structure and vegetation water content change throughout the year in agricultural areas and grasslands [42] and managed grasslands with peat soils using the cross-over angle concept [17]. However, normalization has resulted in only marginal improvement in these studies. Bechtold et al. [15] found that the cross-over angle concept significantly improved the correlation coefficients between the radar backscatter and the observed water table depth; however, this method requires a radar instrument that measures backscatter simultaneously at multiple angles, which is not applicable for Sentinel-1. One promising avenue for making growing vegetation corrections is to combine data from the optical and microwave domains and retrieve the vegetation and soil water states simultaneously [43]. We intend to explore such techniques in future peatland research. To respond to the nature and climate crisis, governments worldwide, including the UK, have set ambitious goals for peatland restoration in the coming years. This in turn has highlighted the need for effective peatland monitoring tools, and earth observation data, such as the satellite imagery from Copernicus programme satellites, have already shown promising results. In this study, we investigated water table depth monitoring possibilities using Sentinel-1 SAR data and three different modelling approaches. Although already promising results were achieved in this study, future studies should investigate if the models' performance is maintained when applied to other peatland sites outside of the study region, and at later stages, if more complex models, additional input variables, and higher resolution SAR data can improve the modelling performance.

## 5. Conclusions

In this study, we investigated the behaviour of Sentinel-1 SAR C-band interaction with Scottish blanket bogs in different conditions. The study focused on the globally rare but highly important blanket bog habitat and aimed to use Sentinel-1 based data for water table depth modelling. To the best of our knowledge, this is the first study to compare the WTD and Sentinel-1 backscatter patterns from near-natural peatlands and sites with different restoration techniques applied.

Three models—SLR, MLR, and RF—using Sentinel-1 radar data were built and tested. The RF model was found to have the highest correlation scores and lowest RMSE values (an $R^2$ of 0.66 for the combined data, and up to 0.66 when used on each site individually). We found that the included categorical covariates, such as the radar image acquisition time (year and season), along with the site identifier and peatland condition group, which have typically not been used in previous studies, can all significantly improve the results of both the MLR and RF models. The impact of standing water, terrain ruggedness, and the ridge and furrow aspect on the model correlation scores was tested, but surprisingly, we found no evidence that these elements had much of an effect on our models. The modelling efforts support the idea that the Sentinel-1 SAR time series data could be used for peatland water table depth monitoring. Given the number of tested sites, it would be beneficial to expand such analysis to further improve the understanding of backscatter–WTD relationships and enhance the precision of the models.

Peatland hydrological dynamics disturbance caused by anthropogenic activities and/or environmental stresses is a threat to peatland conditions and function. These ecosystems require long-term continuous monitoring, and remotely sensed radar data provide the opportunities to both meet the needs of regular continuous monitoring and cover vast

areas. The usage of satellite radar-based data for peatland monitoring is rapidly growing, and, with new radar missions planned for the next decade, it will only increase. Findings from this study may be useful to further improve monitoring and support the management of peatlands.

**Supplementary Materials:** The following supporting information can be downloaded at: https://www.mdpi.com/article/10.3390/rs15071900/s1, Figure S1: Sentinel-1 backscatter time series correction with and without the vegetation sine correction applied.

**Author Contributions:** Conceptualization, all authors; methodology, L.T., R.R.E.A., T.Q., K.M. and A.G.; study design, R.R.E.A., C.S., D.K., M.H.H. and R.H.; formal analysis, L.T., R.R.E.A., T.Q., K.M. and A.G.; software, L.T.; validation, R.R.E.A., T.Q., K.M. and A.G.; investigation, L.T., R.R.E.A., T.Q., K.M. and A.G.; resources, R.R.E.A., C.S., R.H., M.H.H. and D.K.; data curation, L.T. and C.S.; writing—original draft preparation, L.T.; writing—review and editing, all authors; visualization, L.T.; supervision, R.R.E.A., T.Q., K.M. and A.G. All authors have read and agreed to the published version of the manuscript.

**Funding:** L.T. was partly funded by a studentship from The James Hutton Institute, and partly funded by the Natural Environment Research Council (NERC) SCENARIO DTP (Grant Number: NE/L002566/1). R.R.E.A., C.S., and A.G. were supported by the Scottish Government RESAS Strategic Research Programmes (2017-2022 and 2022-2027). T.Q. was supported by the UKRI National Centre for Earth Observation (NE/R016518/1). D.K., M.H.H., and R.H. were funded by the RSPB, with peatland restoration supported by the Scottish Government's Peatland Action funding, the Heritage Lottery Fund, and EU LIFE.

**Data Availability Statement:** The data that support the findings of this study are available from the corresponding author upon reasonable request.

**Acknowledgments:** The authors would like to thank the researchers from James Hutton for the collection and monitoring of the water table depth data in the Forsinard Flows reserve. We would also like to thank the RSPB for site access to their valuable experimental setup.

**Conflicts of Interest:** The authors declare no conflict of interest.

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
