# Peer review of "Potential for Peatland Water Table Depth Monitoring Using Sentinel-1 SAR Backscatter: Case Study of Forsinard Flows, Scotland, UK"

_remotesensing, doi:10.3390/rs15071900_

Round 1

Reviewer 1 Report

This manuscript tries exploring the potential for estimating peatland water table depth (WTD) using Sentinel SAR data, which will be used for monitoring peatland restoration. So, this research is very significant. But, there are some work to be added, which will perfect this research further.

1.       In this research, the three methods are used to fit the relation between the WTD of peatland and the corresponding SAR backscattering, which show the different precisions. The manuscript claims “the previous studies have analyzed the relationship between peatland surface moisture conditions and water table depth with optical, thermal, but less frequently radar satellite data.”(Line 75 and 76). So, the manuscript should show the proposed method based on S1 SAR is different from the previous work based on optical observation or SAR, for example, the difference of the accuracy and application. And, some benchmarking product or method should be regarded as the target. By the aforementioned supplement, the advantage of SAR data estimating peatland WTD will be shown in the manuscript; which is most interesting for the all readers including the journal editor, reviewer and reader.

2.       Although the research adopted the ground observation of WTD in 11 sites, the result based the three models (e.g., SL ML and RF) still be so local. And the manuscript also mentioned the aforementioned problem (Line 43). In the meantime, the modelling result with the better accuracy based on RF wasn’t fully discussed. The aforementioned problems results in an unconvincing conclusion of the broad applicability of the RF method. So, two ways may solve these problems. 1, either adopting more sites for validation; 2, or more profound discussion based on the existing result need to be done, for example, a discussion based on the peatland feature (Line 482- 486) and the radiative transfer theory of RS. The second way may be more interesting.

And, a major revision may be necessary before publication.

Other format problem:

1.       Equation 1 only shown partly in PDF.

2.       Adding the legend for the different results based on the models Figure 3, 4 and 5.

and,due to the limited time,I can't find all problem of gammar and format.

Author Response

We would sincerely like to thank the Reviewer 1 for their time and thoughtful comments and suggestions for improving our manuscript. We have tried our best to improve the manuscript  according to the raised points as much as possible in the short time given. 
Please see the attachment of detailed answers.
Thank you. 

Reviewer 2 Report

In this study,  the application of Sentinel-1 SAR backscatter for water table depth monitoring in near-natural and restored blanket bogs in the Flow Country of northern Scotland is illustrated.  The content of the paper is rich and the conclusion is credible. I have only two questions. 

1. The abstract is a little bit longer.

2. In this paper, the numerical fitting analysis is carried out. Can we clarify the mechanism behind it? The article would be better if it could explain the mechanism.

Author Response

We would sincerely like to thank the Reviewer 2 for their time and suggestions for improving our manuscript. 

Please see point-by-point responses given to comments:

Point 1: The abstract is too long.

Response 1: We have tried our best to shorten the abstract to keep the balance between complete, yet concise message about the paper’s research and findings. It has been shortened from 381 words to 313 words.

Point 2:  In this paper, the numerical fitting analysis is carried out. Can we clarify the mechanism behind it? The article would be better if it could explain the mechanism.

Response 2: In this paper three models with different complexity were fitted: a simple linear regression (SLR), multiple linear regressions (MLR) and random forest method (RF), all conducted in the R programming environment as described in the section 2.5 of the paper. We have added the R commands and packages used for fitting each of the models and given more information about the mechanisms behind each of the models in the manuscript’s section 2.5.

Thank you.

Reviewer 3 Report

remotesensing-2283324
This paper investigates the potential of Sentinel-1 SAR in the monitoring of water table depth of peatland by using of simple linear regression, multiple linear regression and random forest model. The performance of Sentinel-1 SAR over different peatlands including near-natural and restored blanket bogs were assessed. The topic and the experiments are interesting.

The following comments need to be further clarified to ensure the quality of the manuscript.
1 The backscattering of peatland is complex. Although authors have discussed the effect of vegetation on the monitoring, the interactions between vegetation and surface with different water levels are complex and cannot be ignored. It is difficult to correct the scattering caused by vegetation over peatland. It is suggested to give further analysis.
2. There are low correlation coefficients between the backscattering coefficients and WTD in simple linear regression. Different polarimetric channels and categorical variables have been used in Random forest model. It is suggested to compare the importance scores of the inputs derived from random forest model. In addition, it is suggested to compare the results using SAR and environmental data with that using only the VV and VH. The VV and VH may have lower importance scores than that of categorical variables. Please clarify this.
3. Generally, the ridge or terrain ruggedness have large effect on the backscattering of SAR when the ridging aligning direction is perpendicular to the line of sight of radar. The manuscript mentioned that the impact of terrain ruggedness and ridge and furrow aspect on model correlation scores did not have a statistically significant influence. It is suggested to clarify of the aligning direction of ridge or terrain ruggedness.
4. Do the used Sentinel-1 images have same incidence angles? Please clarify this. If the incidence angles were different for the images, it is suggested to give more details about the effect of incidence angles on the temporal profiles of backscattering.
5. Figure 3, 4, and 5 show that there is similar trend between WTD and radar backscattering. But the width of valley of radar backscattering profiles are generally wider than that of WTD. Please clarify this.

Author Response

We would sincerely like to thank the Reviewer 3 for their time and thoughtful comments and suggestions for improving our manuscript. We have tried our best to improve the manuscript  according to the raised points as much as possible in the short time given. 

Please see the attachment of detailed answers.

Thank you.

Round 2

Reviewer 1 Report

that's ok for publishing the revised manuscript, and I hope that we can read the paper of the proposed research by author in the future.

Reviewer 3 Report

The reviwer does not have further comments.